# MCAL: MINIMUM COST HUMAN-MACHINE ACTIVE LABELING

**Hang Qiu[1], Krishna Chintalapudi[2], Ramesh Govindan[1]**
[1]University of Southern California, [2]Microsoft Research
{hangqiu, ramesh}@usc.edu, krchinta@microsoft.com

## ABSTRACT

Today, ground-truth generation uses data sets annotated by cloud-based annotation services. These services rely on human annotation, which can be prohibitively expensive. In this paper, we consider the problem of *hybrid human-machine* labeling, which trains a classifier to accurately auto-label *part* of the data set. However, training the classifier can be expensive too. We propose an iterative approach that minimizes total *overall* cost by, at each step, *jointly* determining which samples to label using humans and which to label using the trained classifier. We validate our approach on well known public data sets such as Fashion-MNIST, CIFAR-10, CIFAR-100, and ImageNet. In some cases, our approach has $6\times$ lower overall cost relative to human labeling the entire data set, and is always cheaper than the cheapest competing strategy.

## 1 INTRODUCTION

Ground-truth is crucial for training and testing ML models. Generating accurate ground-truth was cumbersome until the recent emergence of cloud-based human annotation services (SageMaker (2021); Google (2021); Figure-Eight (2021)). Users of these services submit data sets and receive, in return, annotations on each data item in the data set. Because these services typically employ humans to generate ground-truth, annotation costs can be prohibitively high especially for large data sets.

**Hybrid Human-machine Annotations.** In this paper, we explore using a hybrid human-machine approach to reduce annotation costs (in $) where humans only annotate a subset of the data items; a machine learning model trained on this annotated data annotates the rest. The accuracy of a model trained on a subset of the data set will typically be lower than that of human annotators. However, a user of an annotation service might choose to avail of this trade-off if (a) targeting a slightly lower annotation quality can significantly reduce costs, or (b) the cost of training a model to a higher accuracy is itself prohibitive. Consequently, this paper focuses on the design of *a hybrid human-machine annotation scheme that minimizes the overall cost of annotating the entire data set (including the cost of training the model) while ensuring that the overall annotation accuracy, relative to human annotations, is higher than a pre-specified target (e.g., 95%).*

**Challenges.** In this paper, we consider a specific annotation task, multi-class labeling. We assume that the user of an annotation service provides a set $X$ of data to be labeled and a classifier $\mathcal{D}$ to use for machine labeling. Then, the goal is to find a subset $B \subset X$ human-labeled samples to train $\mathcal{D}$, and use the classifier to label the rest, minimizing total cost while ensuring the target accuracy. A straw man approach might seek to predict human-labeled subset $B$ in a single shot. This is hard to do because it depends on several factors: (a) the classifier architecture and how much accuracy it can achieve, (b) how "hard" the dataset is, (c) the cost of training and labeling, and (d) the target accuracy. Complex models may provide a high accuracy, their training costs may be too high and potentially offset the gains obtained through machine-generated annotations. Some data-points in a dataset are more informative as compared to the rest from a model training perspective. Identifying the "right" data subset for human- vs. machine-labeling can minimize the total labeling cost.

**Approach.** In this paper we propose a novel technique, MCAL[1] (Minimum Cost Active Labeling), that addresses these challenges and is able to minimize annotation cost across diverse data sets. At its

---

[1]MCAL is available at https://github.com/hangqiu/MCAL

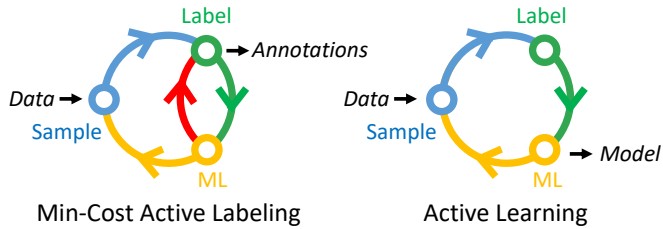

**FIGURE 1:** Differences between MCAL and Active Learning. Active learning outputs an ML model using few samples from the data set. MCAL completely annotates and outputs the dataset. MCAL must also use the ML model to annotate samples reliably (red arrow).

core, MCAL *learns* on-the-fly an *accuracy model* that, given a number of samples to human-label, and a number to machine label, can predict the overall accuracy of the resultant set of labeled samples. Intuitively, this model implicitly captures the complexity of the classifier and the data set. It also uses a *cost model* for training and labeling.

MCAL proceeds iteratively. At each step, it uses the accuracy model and the cost model to *search for the best combination* of the number of samples to human-label and to machine-label that would minimize total cost. It obtains the human-labels, trains the classifier $\mathcal{D}$ with these, dynamically updates the accuracy model, and machine-labels un-labeled samples if it determines that additional training cannot further reduce cost.

MCAL resembles active learning in determining which samples in the data set to select for human-labeling. However, it differs from active learning (Fig. 1) in its goals; active learning seeks to train a classifier with a given target accuracy, while MCAL attempts to label a complete data set within a given error bound. In addition, active learning does not consider training costs, as MCAL does.

This paper makes the following contributions:

• It casts the *minimum-cost labeling* problem in an optimization framework (§2) that minimizes total cost by *jointly selecting* which samples to human label and which to machine label. This framing requires a cost model and an accuracy model, as discussed above (§3). For the former, MCAL assumes that total training cost at each step is proportional to training set size (and derives the cost model parameters using profiling on real hardware). For the latter, MCAL leverages the growing body of literature suggesting that a truncated power-law governs the relationship between model error and training set size (Cho et al. (2015); Hestness et al. (2017); Sala (2019)).

• The MCAL algorithm (§4) refines the power-law parameters, then does a *fast search* for the combination of human- and machine-labeled samples that minimizes the total cost. MCAL uses an active learning metric to select samples to human-label. But because it includes a machine-labeling step, not all metrics work well for MCAL. Specifically, core-set based sample selection is not the best choice for MCAL; the resulting classifier machine-labels fewer samples.

• MCAL extends easily to the case where the user supplies multiple candidate architectures for the classifier. It trains each classifier up to the point where it is able to confidently predict which architecture can achieve the lowest overall cost.

Evaluations (§5) on various popular benchmark data sets show that MCAL achieves lower than the lowest-cost labeling achieved by an oracle active learning strategy. It automatically adapts its strategy to match the complexity of the data set. For example, it labels most of the Fashion data set using a trained classifier. At the other end, it chooses to label CIFAR-100 mostly using humans, since it estimates training costs to be prohibitive. Finally, it labels a little over half of CIFAR-10 using a classifier. MCAL is up to $6\times$ cheaper compared to human labeling all images. It is able to achieve these savings, in part, by carefully determining the training size while accounting for training costs; cost savings due to active learning range from 20-32% for Fashion and CIFAR-10.

## 2 PROBLEM FORMULATION

In this section, we formalize the intuitions presented in §1. The input to MCAL is an unlabeled data set $X$ and a target error rate bound $\varepsilon$. Suppose that MCAL trains a classifier $\mathcal{D}(B)$ using human generated labels for some $B \subset X$. Let the error rate of the classifier $\mathcal{D}(B)$ over the remaining unlabeled data

using $\mathcal{D}(B)$ be $\epsilon(X \setminus B)$. If $\mathcal{D}(B)$ machine-generated[2] labels for this remaining data, the overall ground-truth error rate for $X$ would be, $\epsilon(X) = (1 - |B|/|X|)\,\epsilon(X \setminus B)$. If $\epsilon(X) \geq \varepsilon$, then, this would violate the maximum error rate requirement.

However, $\mathcal{D}(B)$ is still able to generate accurate labels for a carefully chosen subset $S(\mathcal{D}, B) \subset X \setminus B$ (*e.g.,* comprising only those that $\mathcal{D}(B)$ is very confident about). After generating labels for $S(\mathcal{D}, B)$ using $\mathcal{D}(B)$, labels for the remaining $X \setminus B \setminus S(\mathcal{D}, B)$ can be once again generated by humans. The *overall error rate* of the generated ground-truth then would be $\{(|S(\mathcal{D}, B)|)/(|X|)\}\epsilon(S(\mathcal{D}, B))$. $\epsilon(S(\mathcal{D}, B))$ is the error rate of generating labels over $S(\mathcal{D}, B)$ using $\mathcal{D}(B)$ and is, in general, higher for larger $|S(\mathcal{D}, B)|$. Let $S^{\star}(\mathcal{D}, B)$ be the largest possible $S(\mathcal{D}, B)$ that ensures that the overall error rate is less than $\varepsilon$. Then, overall cost of generating labels is:

$$\mathbf{C} = (|X \setminus S^{\star}(\mathcal{D}, B)|) \cdot C_h + C_t(\mathcal{D}(B)) \tag{1}$$

where, $C_h$ is the cost of human labeling for a single data item and $C_t(\mathcal{D}(B))$ is the total cost of generating $\mathcal{D}(B)$ including the cost of optimizing $B$ and training $\mathcal{D}(B)$. Given this, MCAL achieves the minimum cost by *jointly selecting* $S^{\star}(\mathcal{D}, B)$ and $B$ subject to the accuracy constraint:

$$\mathbf{C}^{\star} = \underset{S^{\star}(\mathcal{D}, B), B}{\mathrm{argmin}} \ \mathbf{C}, \ \text{ s.t. } \ (|S^{\star}(\mathcal{D}, B)|)/(|X|)\epsilon(S^{\star}(\mathcal{D}, B)) < \varepsilon \tag{2}$$

## 3 Cost Prediction Models

MCAL must determine the optimal value of the training size $B$ and its corresponding maximum machine-labeled set $S^{\star}(\mathcal{D}, B)$ that minimizes $\mathbf{C}$. In order to make optimal choices, MCAL must be able to predict $\mathbf{C}$ as a function of the choices. $\mathbf{C}$ in turn depends on $|S^{\star}(\mathcal{D}, B)|$ and the training cost $C_t(\mathcal{D}(B))$ (Eqn. 1). Thus, MCAL actually constructs two predictors, one each for $|S^{\star}(\mathcal{D}, B)|$ (§3.1) and $C_t(\mathcal{D}(B))$ (§3.2). These predictors rely on two *sample selection functions* (§3.3): from among the remaining un-labeled samples, $M(.)$ selects, which to human-label for training; $L(.)$ selects those that the classifier $\mathcal{D}(B)$ can machine-label within the error constraint $\varepsilon$.

### 3.1 Estimating Machine-Labeling Performance

To maximize total cost reduction without violating the overall accuracy constraint (Eqn. 1), MCAL must determine a maximal fraction of samples $\theta^{\star}$ selected by $L(.)$ for machine labeling. To predict the model error $\epsilon(S^{\theta}(\mathcal{D}(B)))$, we leverage recent work (§6) that observes that, for many tasks and many models, the generalization error *vs.* training set size is well-modeled by a power-law (Hestness et al. (2017); Sala (2019)) ( $\epsilon(S^{\theta}(\mathcal{D}(B))) = \alpha_{\theta}\,|B|^{-\gamma_{\theta}}$ ). However, it is well-known that most power-laws experience a fall-off (Burroughs (2001)) at high values of the independent variable. To model this, we use an ***upper-truncated*** power-law (Burroughs (2001)):

$$\epsilon(S^{\theta}(\mathcal{D}(B))) = \alpha_{\theta}\,|B|^{-\gamma_{\theta}}\, e^{-\frac{|B|}{k_{\theta}}} \tag{3}$$

where $\alpha_{\theta}, \gamma_{\theta}, k_{\theta}$ are the power-law parameters.

Eqn. 3 can better predict the generalization error (Fig. 2) based on the training size than a power-law, especially at larger values of $|B|$. Fig. 3 shows that more samples help calibrate the truncated power law to better predict the falloff. Other models and data sets show similar results (see Appendix §F).

$\epsilon(S^{\theta}(\mathcal{D}(B)))$ will increase monotonically with $\theta$ since increasing $\theta$ has the effect of adding data that $\mathcal{D}$ is progressively less confident about (ranked by $L(.)$). Lacking a parametric model for this dependence, to find $\theta^{\star}$, MCAL generates power-law models $\epsilon(S^{\theta}(\mathcal{D}(B)))$ for various discrete $\theta \in (0, 1)$ (§4). MCAL obtains $\theta^{\star}$ for a given $B$ by searching across the predicted $\epsilon(S^{\theta}(\mathcal{D}(B)))$. While $\epsilon(S^{\theta}(\mathcal{D}(B)))$ also depends on the data acquisition batch size ($\delta$, see §3.2), when the total accumulated training sample size is large enough, the error rate starts to converge, hence the dependence is insignificant. Fig. 4 shows this variation is $< 1\%$ especially for smaller values of $\theta$.

### 3.2 Estimating Active Learning Training Costs

Active learning iteratively obtains human labels for a batch size of $\delta$ items ranked using the sample selection function $M(.)$ and adds them to the training set $B$ to retrain the classifier $\mathcal{D}$. A smaller

---

[2]Evaluation of mixed-labels assumes perfect human-labels($B$).

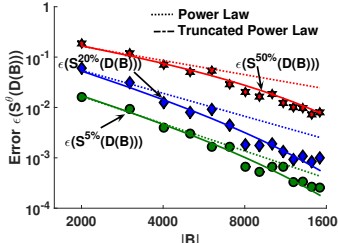 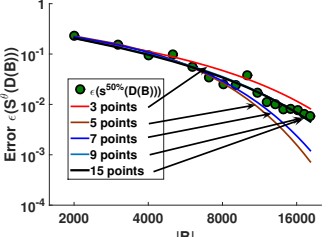 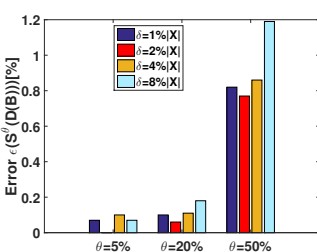

**FIGURE 2:** Fitting generalization error ($\epsilon(S^\theta(\mathcal{D}(B)))$) agasint training size to a power law and a truncated power law (CIFAR-10 using RESNET18 for various $\theta$).

**FIGURE 3:** Error prediction improves with increasing number of error estimates for CIFAR-10 using RESNET18.

**FIGURE 4:** Dependence of $\epsilon(S^\theta(\mathcal{D}(B)))$ on $\delta$ is "small" especially towards the end of active learning. Here, ($|B| = 16,000$) for CIFAR-10 using RESNET18.

$\delta$ typically makes the active learning more effective: it allows for achieving a lower error for a potentially smaller $B$ through more frequent sampling, but also significantly increases the training cost due to frequent re-training. Choosing an appropriate training sample acquisition batch size $\delta$ is thus an important aspect of minimizing overall cost.

The training cost depends on the training time, which in turn is proportional to the data size ($|B|$) and the number of epochs. A common strategy is to use a fixed number of epochs per iteration, so the training cost in each iteration is proportional to $|B|$. Assuming $\delta$ new data samples added in each iteration, the total training cost follows

$$C_t(\mathcal{D}(B)) = \frac{1}{2}|B|(|B|/\delta + 1) \tag{4}$$

MCAL can accommodate other training cost models[3].

## 3.3 SAMPLE SELECTION

The sample selection functions $M(.)$ and $L(.)$ are qualitatively different. The former selects samples to obtain an accurate classifier, the latter determines which samples a given classifier is most confident about (so they can be machine-labeled). For $L(.)$, we simply use the *margin* metric (Scheffer et al. (2001)), the score difference between the highest and second highest ranked labels (see Fig. 5).

$M(.)$ is similar to sample selection for active learning. Prior work has considered coreset (Sener & Savarese (2017)) and uncertainty (Scheffer et al. (2001)) based approaches. The $k$-center selection method (Wolf (2011)), an instance of the former, uses the feature-space distance between the last activation layer output to iteratively select data points farthest from existing centers.

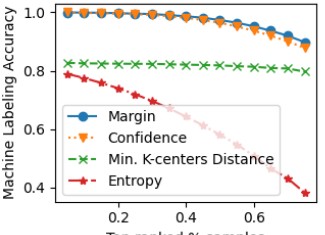 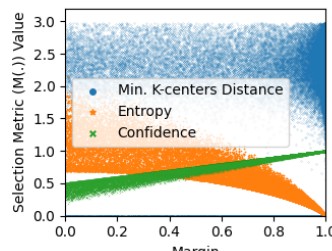

**FIGURE 5:** Machine labeling accuracy of samples ranked by $L(.)$

**FIGURE 6:** Sampling metric ($M(.)$) comparison

Uncertainty-based approaches include *max-entropy* (Dagan & Engelson (1995)), which samples data with highest entropy in activation output, *least-confidence* (Culotta & McCallum (2005)), and margin.

We evaluate MCAL using these different choices for $M(.)$, but find that (§5), for a subtle reason, uncertainty-based approaches perform better. Unlike with active learning, where the goal is to select samples to match the generalization error of the entire data set, MCAL's goal is to select samples to train a classifier that can machine label the largest number of samples possible. When MCAL uses margin or least confidence for $L(.)$, the samples selected have high accuracy (close to 100%,

---

[3]For example, if the number of epochs is proportional to $|B|$ in which case $C_t(\mathcal{D}(B))$ can have a cubic dependency on $|B|$

Fig. 5)[4]. The samples selected by a core-set based algorithm such as $k$-centers is poorly correlated with accuracy (Fig. 5) and margin (Fig. 6).

# 4    THE MCAL ALGORITHM

The MCAL algorithm (Alg. 1, see appendix §A) takes as input an active learning metric $M(.)$, the data set $\mathbf{X}$, the classifier $\mathcal{D}$ (*e.g.,* RESNET18) and parametric models for training cost (*e.g.,* Eqn. 4), and for error rate as a function of training size (*e.g.,* the truncated power law in  Eqn. 3). The algorithm operates in two phases. In the first phase, it uses estimates obtained during active learning to learn the parameters of one truncated power-law model for each discrete machine-label fraction $\theta$, and uses the cost measurements to learn the parameters for the training-cost model. In the second phase, having the models, it can estimate and refine the optimal machine-labeling subset $S^\star(\mathcal{D}, B)$ and the corresponding training set $B$ that produce the optimal cost $\mathbf{C}^\star$. It can also estimate the optimal batch size $\delta_{opt}$ for this cost. It terminates when adding more samples to $B$ is counter productive. It trains a classifier to label $S^\star(\mathcal{D}, B)$ and use human labels for the remaining unlabeled samples.

To start with, MCAL randomly selects a test set $\mathbf{T}$ (*e.g.,* $|T| = 5\%$ of $|X|$) and obtains human labels to test and measure the performance of $\mathcal{D}$. Next, MCAL initializes $B = B_0$ by randomly selecting $\delta_0$ samples from $X$ (*e.g.,* 1% of $X$ in our implementation) and obtaining human labels for these. Then, MCAL trains $\mathcal{D}$ using $B_0$ and uses $T$ to estimate the generalization errors $\epsilon_T\left(S^\theta(\mathcal{D}(B_0))\right)$ for various values of $\theta \in (0, 1)$ (we chose in increments of 0.05 $\{0.05, 0.1, \cdots, 1\}$), using $T$ and $M(.)$.

Next, the main loop of MCAL begins. Similar with active learning, MCAL selects, in each step, $\delta$ samples, ranked by $M(.)$, obtains their labels and adds them to $B$, then trains $\mathcal{D}$ on them. The primary difference with active learning is that MCAL, in every iteration, estimates the model parameters for machine-labeling error $\epsilon(S^\theta(\mathcal{D}(B)))$ for each $\theta$ with various training size $B$, then uses these to estimate the best combination of $B$ and $\theta$ that will achieve the lowest corresponding overall cost $\mathbf{C}^\star$. At the end of this iteration, MCAL can answer the question: "How many human generated labels must be obtained into $B$ to train $D$, in order to minimize $\mathbf{C}$?" (§3).

The estimated model parameters for the training cost $C_t(\mathcal{D}(B))$ and the machine-label error $\epsilon(S^\theta(\mathcal{D}(B)))$ may not be stable in the first few iterations given limited data for the fit. To determine if the model parameters are stable, MCAL compares the estimated $\mathbf{C}^\star$ obtained from the previous iteration to the current. If the difference is small ($\leq 5\%$, in our implementation), the model is considered to be stable for use.

After the predictive models have stabilized, we can rely on the estimates of the optimal training size $B_{opt}$, to calculate the final number of labels to be obtained into $B$. At this point MCAL adjusts $\delta$ to reduce the training cost when it is possible to do so. MCAL can do this because it targets relatively high accuracy for $\mathcal{D}(B)$. For these high targets, it is important to continue to improve model parameter estimates (*e.g.,* the parameters for the truncated power law), and active learning can help achieve this. Fig. 3 shows how the fit to the truncated power law improves as more points are added. Finally, unlike active learning, MCAL adapts $\delta$, proceeding faster to $B_{opt}$ to reduce training cost. Fig. 4 shows that, for most values of $\theta$, the choice of $\delta$ does not affect the final classifier accuracy much. However, it can significantly impact training cost (§3).

This loop terminates when total cost obtained in a step is higher than that obtained in the previous step. At this point, MCAL simply trains the classifier using the last predicted optimal training size $B_{opt}$, then human labels any remaining unlabeled samples.

**Extending MCAL to selecting the cheapest DNN architecture.** In what we have described so far, we have assumed that MCAL is given a candidate classifier $D$. However, it is trivial to extend MCAL to the case when the data set curator supplies a small number (typically 2-4) of candidate classifiers $\{\mathcal{D}_1, \mathcal{D}_2, \cdots\}$. In this case, MCAL can generate separate prediction models for each of the classifiers and pick the one that minimizes $\mathbf{C}$ once the model parameters have stabilized. This does not inflate the cost significantly since the training costs until this time are over small sizes of $B$.

**Accommodating a budget constraint.** Instead of a constraint of labeling error, MCAL can be modified to accommodate other constraints, such as a limited total budget. Its algorithm can search

---

[4]The results are from Res18 trained over 8K samples (CIFAR10); actual numbers will vary by dataset, training size, and model used.

for the lowest error satisfying a given budget constraint. Specifically, with the same model for estimation of network error and total cost (§3), instead of searching the $S^\star(\mathcal{D}, B)$ for minimum total cost while error constraint is satisfied (line 18 of Alg. 1, Appendix §A), we can search for minimum estimated error while the total cost is within budget. The difference is: in the former case, we can always resort to human labeling when error constraint cannot be satisfied; in the latter case, we can only sacrifice the total accuracy by stopping the training process and taking the model's output when the money budget is too low.

## 5 EVALUATION

In this section, we evaluate the performance of MCAL over popular classification data sets: Fashion-MNIST, CIFAR-10, CIFAR-100, and ImageNet. We chose these data sets to demonstrate that MCAL can work effectively across different difficulty levels, We use three popular DNN architectures RESNET50, RESNET18 (He et al. (2016)), and CNN18 (RESNET18 without the skip connections). These architectures span the range of architectural complexity with differing training costs and achievable accuracy. This allows us to demonstrate how MCAL can effectively select the most cost efficient architecture among available choices. We also use two different labeling services: Amazon SageMaker (2021) at \$0.04/image and Satyam (Qiu et al. (2018)) at \$0.003/image. This allows us to demonstrate how MCAL adapts to changing human labeling costs. Finally, parametric model fitting costs are negligible compared to training, metric profiling, and human labeling costs.

MCAL evaluates different sampling methods discussed (Fig. 6) and uses margin to rank and select samples. At each active learning iteration, it trains the model over 200 epochs with a $10\times$ learning rate reduction at 80, 120, 160, 180 epochs, and a mini-batch-size of 256 samples (Keras (2021)). Training is performed on virtual machines with 4 NVIDIA K80 GPUs at a cost of 3.6 USD/hr. In all experiments, unless otherwise specified, the labeling accuracy requirement $\varepsilon$ was set at 5%.

### 5.1 REDUCTION IN LABELING COSTS USING MCAL

MCAL automatically makes three key decisions to minimize overall labeling cost. It a) selects the subset of images that the classifier should be trained on ($|B|_{opt}$), b) adapts $\delta$ across active learning iterations to keep training costs in check, and c) selects the best DNN architecture from among a set of candidates. In this section, we demonstrate that MCAL provides significant overall cost benefits at the expense of $\varepsilon$ (5%) degradation in label quality. Further, it even outperforms active learning assisted by an oracle with optimal $\delta$ value.

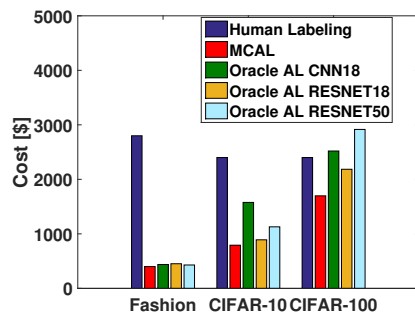

**FIGURE 7:** Total cost of labeling for various data sets, for i) Human labeling, ii) MCAL, and iii) Oracle assisted AL with various DNN architectures.

Fig. 7 depicts the total labeling costs incurred when using Amazon labeling services for three different schemes: i) when the entire data set is labeled by Amazon labeling services, ii) MCAL for $\varepsilon = 5\%$, and iii) active learning with an oracle to choose $\delta$ for each DNN model.

Tbl. 1 lists the numerical values of the costs (in \$) for human labeling and MCAL. To calculate the total labeling error, we compare the machine labeling results on $S^\star(\mathcal{D}, B_{opt})$ and human labeling results on $X \setminus S^\star(\mathcal{D}, B_{opt})$ against the ground-truth. The human labeling costs are calculated based on the prices of Amazon labeling services SageMaker (2021) and Satyam (Qiu et al. (2018)).

**Cost Saving Compared to Human Labeling.** From Fig. 7 and Tbl. 1, MCAL provides an overall cost saving of 86%, 67% and 30% for Fashion, CIFAR-10 and CIFAR-100 respectively. As expected, the savings depend on the difficulty of the classification task. The "harder" the dataset the higher the savings. Tbl. 1 also shows the number of samples in $B$ used to train $\mathcal{D}$, as well as the number of samples $|S|$ labeled using $\mathcal{D}$. For Fashion, MCAL labels only 6.1% of the data to train the classifier and uses it to label 85% of the data set. For CIFAR-10, it trains using 22% of the data set and labels about 65% of the data using the classifier. CIFAR-100 requires more data to train the classifier to a high accuracy so is able to label only 10% of the data using the classifier.

| Data Set | Labeling Service | $\frac{|B|}{|X|}$ | $\frac{|S|}{|X|}$ | DNN Selected | Error | Human Cost ($) | MCAL Cost ($) | MCAL Savings |
|---|---|---|---|---|---|---|---|---|
| Fashion | Amazon | 6.1% | 85.0% | Res18 | 4.0% | 2800 | 400 | 86% |
| | Satyam | 8.4% | 85.0% | Res18 | 4.0% | 210 | 29 | 86% |
| CIFAR 10 | Amazon | 22.2% | 65.0% | Res18 | 2.4% | 2400 | 792 | 67% |
| | Satyam | 27.0% | 65.0% | Res18 | 2.4% | 180 | 63 | 65% |
| CIFAR 100 | Amazon | 32.0% | 10.0% | Res18 | 0.4% | 2400 | 1698 | 29% |
| | Satyam | 57.6% | 20.0% | Res18 | 1% | 180 | 139 | 23% |

**TABLE 1:** Summary of Results

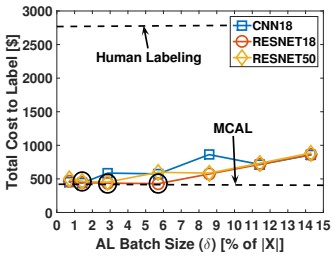

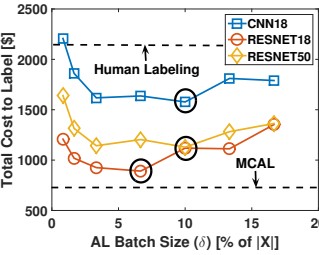

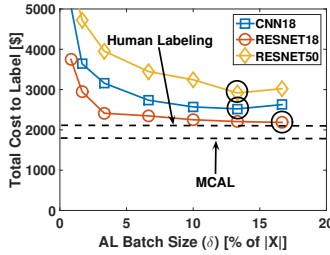

**FIGURE 8:** MCAL vs. active learning with different batch sizes $\delta$ (Fashion, Amazon).

**FIGURE 9:** MCAL vs. active learning with different batch sizes $\delta$ (CIFAR-10, Amazon).

**FIGURE 10:** MCAL vs. active learning with different batch sizes $\delta$ (CIFAR-100, Amazon).

**Cost Savings from Active Learning.** In the absence of MCAL, how much cost savings would one obtain using naive Active Learning? As described in §3, the overall cost of AL depends on the batch size $\delta$, the DNN architecture used for classification, as well as how "hard" it is to classify the data set. In order to examine these dependencies, we performed active learning using values of $\delta$ between 1% to 20% of $|\mathbf{X}|$, to label each data set using the different DNN architectures until the desired overall labeling error constraint was met. The contribution to overall labeling error is zero for human annotated images and dictated by the classifier performance for machine labeled images.

**MCAL v.s. AL.** Figures 8, 9 and 10 show the overall cost for each of the three data sets using different DNN architectures. The optimal value of $\delta$ is indicated by a circle in each of the figures. Further, the cost of labeling using humans only as well as MCAL cost is indicated using dashed lines for reference. As shown in the figure, MCAL outperforms AL even with optimal choice of $\delta$. The choice of $\delta$ can significantly affect the overall cost, by up to 4-5× for hard data sets.

**Training cost.** Figures 19, 20, and 21 (see appendix §E) depict the AL training costs for each of the data sets using three different DNNs. The training cost can vary significantly with $\delta$. While for Fashion, there is a 2× reduction in training costs, it is about 5× for CIFAR-10 and CIFAR-100.

**Dependence on sample selection method $M(.)$.** Fig. 11 shows the total cost of MCAL using different sample selection functions $M(.)$. As §3.3 describes, *k-center* is qualitatively different from uncertainty-based methods in the way it selects samples. In Fig. 11, for CIFAR10 on RESNET18 (other results are similar, omitted for brevity), while all metrics reduce cost relative to human-labeling the entire data set, uncertainty-based metrics have 25% lower cost compared to $k$-center because the latter machine-labels fewer samples. Thus, for active labeling, an uncertainty-based metric is better for selecting samples for classifier training.

**Dependence on batch size $\delta$.** Fig. 12 depicts the fraction of images that were machine labeled for the different data sets and DNN architectures trained with different batch sizes. As seen in Fig. 12, lower $\delta$ values allow AL to adapt at a finer granularity, resulting in a higher number of machine labeled images. Increasing $\delta$ from 1% to 15% results in a 10-15% fewer images being machine labeled.

**Dependence on data set complexity.** As seen from Figures 8-10, and 12, the training costs as well as potential gains from machine labeling depend on the complexity of the data set. While, using a small $\delta$ (2%) is beneficial for Fashion, the number is larger for CIFAR-10. For CIFAR-100, using active learning does not help at all as the high training costs overwhelm benefits from machine labeling.

A second dimension of complexity is the number of samples per class. CIFAR-100 has 600 per class, CIFAR-10 has 6000. To demonstrate that MCAL adapts to varying numbers of samples per class, Fig. 13 shows an experiment in which we ran MCAL on subsets of CIFAR-10, with $1000-5000$

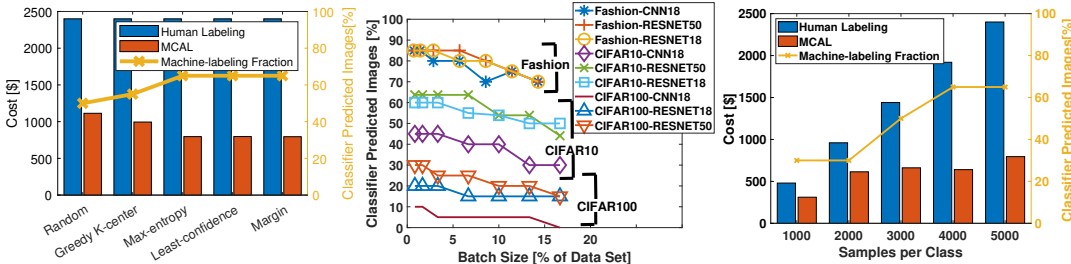

**FIGURE 11:** Total cost and machine-labeling fraction of MCAL using different sampling methods (RESNET18, CIFAR-10)

**FIGURE 12:** Fraction of machine labeled images ($|S^*(D(B))|/|X|$) using Naive AL for different $\delta$.

**FIGURE 13:** Total cost and machine-labeling fraction of MCAL using RESNET18 on various subset sizes of CIFAR-10

| Data Set | Labeling Service | DNN Architecture | | | | | | | | |
|---|---|---|---|---|---|---|---|---|---|
| | | CNN-18 | | | RESNET18 | | | RESNET50 | | |
| | | $\delta_{opt}$ | Cost ($) | Savings | $\delta_{opt}$ | Cost ($) | Savings | $\delta_{opt}$ | Cost ($) | Savings |
| Fashion | Amazon | 1.7% | 438.37 | 84.3% | 6.7% | 429.27 | 84.6% | 3.3% | 452.36 | 83.8% |
| | Satyam | 1.7% | 49.87 | 76.3% | 6.7% | 40.77 | 75.3% | 3.3% | 63.86 | 69.6% |
| CIFAR 10 | Amazon | 10% | 1577.24 | 34.3% | 6.7% | 891.06 | 62.9% | 10% | 1128.65 | 53.0% |
| | Satyam | 16.7% | 235.81 | -31.0% | 6.7% | 129.60 | 28.0% | 10% | 149.63 | 16.9% |
| CIFAR 100 | Amazon | 13.3% | 2520.79 | -5% | 16.7% | 2184.60 | 9.0% | 13.3% | 2915.80 | -21.5% |
| | Satyam | 16.7% | 407.04 | -126.1% | 16.7% | 297.60 | -65.3% | 16.7% | 805.76 | -347.6% |

**TABLE 2:** Oracle Assisted Active Learning

randomly selected samples per class. With 1000 samples per class, the majority of samples were used for training so MCAL can machine-label only 30%. With 5000 samples per class, this fraction goes up to 65%, resulting in increased cost savings relative to human-labeling the entire data set.

***Dependence on DNN architecture.*** While a larger DNN has the potential for higher accuracy, its training costs may be significantly higher and may potentially offset savings due to machine labeling. As seen from Figures 8–10, and 12, even though RESNET50 is able to achieve a higher prediction quality and machine-labels more images, its high training cost offsets these gains. CNN18 on the other hand incurs much lower training costs, however, its poor performance leads to few images being machine-labeled. RESNET18 provides for a better compromise resulting in overall lower cost.

**MCAL on Imagenet.** We have explored applying MCAL on Imagenet using an EfficientNetB0 (Tan & Le (2019)). Relative to CIFAR-10 data set, Imagenet (with over 1.2 M images) is challenging for MCAL because: it has more classes (1000), fewer samples per class (avg. 1200), and the training cost is 60-200× higher than that for RESNET-18. For these reasons, MCAL trains the network to over 80% accuracy up to 454K images, and decides, because it cannot machine label any, to human-label the entire data set. As with CIFAR-100, for complicated data sets, MCAL still makes the correct decision.[5]

**Summary.** Tbl. 2 provides the optimal choices for $\delta$ and the optimal cost savings obtained for various DNNs. Comparing these values with Tbl. 1, we conclude that: MCAL outperforms Naive AL across various choices of DNN architectures and $\delta$ by automatically picking the right architecture, adapting $\delta$ suitably, and selecting the right subset of images to be human labeled.

## 5.2 GAINS FROM ACTIVE LEARNING

Figures 14 and 15 ( see appendix §B) show the overall labeling cost with and without AL for the three data sets using Amazon and Satyam labeling services. The percentage cost gains are in brackets. While Fashion and CIFAR-10 show a gain of about 20% for both data sets, the gains are low in the case of CIFAR-100 because most of the images in that data set were labeled by humans and active learning did not have an opportunity to improve significantly. The gains are higher for Satyam, since training costs are relatively higher in that cost model: active learning accounted for 25-31% for Fashion and CIFAR-10's costs, and even CIFAR-100 benefited from this.

---

[5]To make this decision, MCAL terminates when it has spent more than $x\%$ ($x = 10$ in our evaluations) of the human labeling cost to train the classifier before reaching the desired accuracy. So, for any highly complex data set for which it decides to human-label all samples, MCAL pays a small "exploration tax" before termination.

### 5.3 Effect of Relaxing Other Assumptions

**Cheaper Labeling Cost.** Intuitively, with cheaper labeling costs MCAL should use more human labeling to train the classifier. This in turn should enable a larger fraction of data to be labeled by the classifier. To validate this, we used the Satyam (Qiu et al. (2018)) labeling service (with $10\times$ lower labeling cost). The effect of this reduction is most evident for CIFAR-100 in Tbl. 1 as MCAL chooses to train the classifier using 57.6% of the data (instead of 32% using Amazon labeling service). This increases the classifier's accuracy allowing it to machine-label 10% more of the data set. For other data sets, the differences are less dramatic (they use 2.5-5% more data to train the classifier, more details in appendix §C, Figures 16, 17, and 18). The numerical values of the optimal $\delta$ as well as the corresponding cost savings are provided in Table 2: MCAL achieves a lower overall cost compared to all these possible choices in this case as well.

**Relaxing Accuracy Requirement.** In appendix §D, we examine the quality-cost trade-off by relaxing the accuracy target from 95% to 90% to quantify its impact on additional cost savings. Fashion achieves 30% cost reductions; many more images are labeled by the classifier. CIFAR-10 and CIFAR-100 also show 10-15% gains.

## 6 Related Work

Active learning (Settles (2010)) aims to reduce labeling cost in training a model, by iteratively selecting the most informative or representative samples for labeling. Early work focused on designing metrics for sample selection based on coreset selection (Sener & Savarese (2017)), margin sampling (Scheffer et al. (2001); Jiang & Gupta (2019)), region-based sampling (Cortes et al. (2020)), max entropy (Dagan & Engelson (1995)) and least confidence (Culotta & McCallum (2005)). Recent work has focused on developing metrics tailored to specific tasks, such as classification (Coleman et al. (2020)), detection (Brust et al. (2019)), and segmentation (Yang et al. (2017)), or for specialized settings such as when costs depend upon the label (Krishnamurthy et al. (2017)), or for a hierarchy of labels (Hu et al. (2019)). Other work in this area has explored variants of the problem of sample selection: leveraging selection-via-proxy models (Coleman et al. (2020)), model assertions (Kang et al. (2020)), model structure (Wang et al. (2017)), using model ensembles to improve sampling efficacy (Beluch et al. (2018)), incorporating both uncertainty and diversity (Ash et al. (2020)), or using self-supervised mining of samples for active learning to avoid data set skew (Wang et al. (2018)). MCAL uses active learning and can accommodate multiple sample selection metrics.

Training cost figures prominently in the literature on hyper-parameter tuning, especially for architecture search. Prior work has attempted to predict learning curves to prune hyper-parameter search (Klein et al. (2017)), develop effective search strategy within a given budget (Lu et al. (2019)), or build a model to characterize maximum achievable accuracy to enable fast triage during architecture search (Istrate et al. (2019)), all with the goal of reducing training cost. The literature on active learning has recognized the high cost of training for large datasets and has explored using cheaper models (Coleman et al. (2020)) to select samples. MCAL solves a different problem (dataset labeling) and explicitly incorporates training cost in reasoning about which samples to select.

Also relevant is the empirical work that has observed a power-law relationship between generalization error and training set size (Beery et al. (2018); Johnson et al. (2018); Figueroa et al. (2012)) across a wide variety of tasks and models. MCAL builds upon this observation, and learns the parameters of a truncated power-law model with as few samples as possible.

## 7 Conclusions

Motivated by the need of data engineering for increasingly advanced ML applications as well as the the prohibitive human labeling cost, this paper asks: "How to label a data set at minimum cost"? To do this, it trains a classifier using a set $B$ from the data set to label $S$ samples, and uses humans to label the rest. The key challenge is to balance human- ($B$) vs. machine-labels ($S$) that minimizes total cost. MCAL jointly optimizes the selection by modeling the active learning accuracy vs. training size as a truncated power-law, and search for minimum cost strategy while respecting error constraints. The evaluation shows that it can achieve up to $6\times$ lower cost than using humans, and is always cheaper than active learning with the lowest-cost batch size.

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

## A  MCAL ALGORITHM

---

**Algorithm 1** MCAL

---

**Inputs:** An active learning metric $M(.)$, a classifier $\mathcal{D}$, set of unlabeled images $(X)$, a parametric model to predict $C_t(\mathcal{D}(B))$ and a parametric model to predict $\epsilon(S^\theta(\mathcal{D}(B)))$

1: Obtain human generated labels for a randomly sampled test set $T \subset X$, and let $X = X \setminus T$.
2: Obtain human generated labels for a randomly sampled data items $B_0 \subset X$, $|B_0| = \delta_0$.
3: Train $\mathcal{D}(B_0)$ and test the classifier over $T$
4: Record training cost $C_t(\mathcal{D}(B_0))$
5: **for** $\theta \in \{\theta_{min}, \ldots, \theta_{max}\}$ **do**
6:    Estimate $\epsilon_T\left(S^\theta(\mathcal{D}(B_0))\right)$ using $T$ and $M(.)$
7: **end for**
8: Initialization: $\mathbf{C}^\star_{new} = 0, \mathbf{C}^\star_{old} = 0, \delta = \delta_0, i = 1, B_{opt} = B_0$
9: **while** $\mathbf{C}^\star < \mathbf{C}(B_{opt} + \delta)$ **do**
10:    Obtain human generated labels for $b_i \subset X \setminus B_{i-1}$ comprising $|b_i| = \delta$ samples ranked using $M(.)$
11:    $B_i = B_{i-1} \cup b_i$
12:    Train $\mathcal{D}(B_i)$ and test the classifier over $T$
13:    Record $C_t(\mathcal{D}(B_i))$ and estimate $C_t(\mathcal{D}(B))$ using $\langle|B_k|, C_t(\mathcal{D}(B_k))\rangle, \forall k$
14:    **for** $\theta \in [\theta_{min}, \cdots, \theta_{max}]$ **do**
15:       Estimate $\epsilon_T\left(S^\theta(\mathcal{D}(B_i))\right)$ using $T$ and $M(.)$
16:       Estimate and update the error model parameters $(\alpha_\theta, \gamma_\theta, k_\theta$ from Eqn. 3) for $\epsilon(S^\theta(\mathcal{D}(B)))$ using $\langle|B_k|, \epsilon_T\left(S^\theta(\mathcal{D}(B_k))\right)\rangle, \forall k$
17:    **end for**
18:    Find $\mathbf{C}^\star_{new} = \mathbf{C}^\star$, $B_{opt}$ as described in Section 3
19:    **if** $(|\mathbf{C}^\star_{new} - \mathbf{C}^\star_{old}|)/|\mathbf{C}^\star_{new}| < \Delta$ **then**
20:       $\arg\min_N \delta_{opt} = (|\mathbf{B}_{opt}| - |\mathbf{B}_i|)/N, s.t. \mathbf{C} < \mathbf{C}^\star(1 + \beta)$
21:       $\delta = \delta_{opt}$
22:    **end if**
23:    $\mathbf{C}^\star_{old} = \mathbf{C}^\star_{new}$
24:    $i = i + 1$
25: **end while**
26: Use $\mathcal{D}(B_{opt})$ and $L(.)$ to find $S^\star(\mathcal{D}, B_{opt})$
27: Annotate the residual $X \setminus B \setminus S^\star(\mathcal{D}, B_{opt})$

---

## B  COST SAVINGS ON ACTIVE LEARNING

Figures 14 and 15 show the overall labeling cost with and without AL for the three data sets using Amazon and Satyam respectively The percentage cost gains are in brackets. While Fashion and CIFAR-10 show a gain of about 20% for both data sets, the gains are low in the case of CIFAR-100 because most of the images in that data set were labeled by humans and active learning did not have an opportunity to improve significantly. The gains are higher for Satyam, since training costs are relatively higher in that cost model: active learning accounted for 25-31% for Fashion and CIFAR-10's costs, and even CIFAR-100 benefited from this.

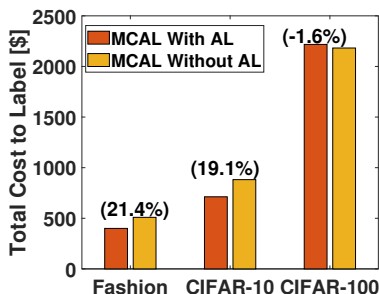

**FIGURE 14:** Active learning gains using Amazon Labeling

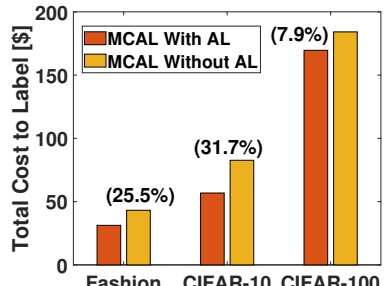

**FIGURE 15:** Active learning gains using Satyam Labeling

## C   RESULTS ON CHEAPER LABELING SERVICE

Figures 16, 17, and 18 depict the effect of using Satyam as the labeling service. As seen in these figures, the lower labeling cost alters the tradeoff curves. The figures also depict the corresponding MCAL cost as well as the human labeling cost for reference. MCAL achieves a lower overall cost compared to all these possible choices in this case as well.

## D   EFFECT OF RELAXING ACCURACY REQUIREMENT

Tbl. 3 depicts the fraction of images that were machine-labeled by the classifier ($|\mathbf{S}|/|\mathbf{X}|$) and the number of samples used to train it ($|\mathbf{B}|/|\mathbf{X}|$) for each of the data sets. Comparing with Tbl. 1, for Fashion MCAL predicts more images by using a smaller number of training images. For CIFAR-10 and CIFAR-100, it makes a different decision and uses more training images to increase the classifier accuracy to enable more images to be machine-labeled. Further, RESNET18 continues to be the optimal architecture for all three data sets. As seen from Tbl. 3, MCAL ensures the accuracy target of 90% for all the data sets. Tbl. 3 also captures savings with respect to human labeling while using an accuracy guarantee of 90%. However, the savings do not dramatically increase indicating that most of the cost gain comes in reducing the accuracy requirement to 95% from 100%.

| Data Set | $\frac{|\mathbf{B}|}{|\mathbf{X}|}$ | $\frac{|\mathbf{S}|}{|\mathbf{X}|}$ | DNN Selected | Labeling Accuracy | Cost Savings |
|---|---|---|---|---|---|
| Fashion | 4.4% | 90.0% | RES18 | 91.9% | 88.9% |
| CIFAR-10 | 25.9% | 75.0% | RES18 | 94.7% | 70.5% |
| CIFAR-100 | 64.0% | 25.0% | RES18 | 98.4% | 39.1% |

**TABLE 3:** Relaxing Error Constraints to $\varepsilon = 10\%$

## E   TRAINING COST

Fig. 19, Fig. 20, Fig. 21 shows the training cost portion from the total cost (Fig. 8,Fig. 9, Fig. 10). The training cost can vary significantly with $\delta$. While for Fashion, there is a $2\times$ reduction in training costs, it is about $5\times$ for CIFAR-10 and CIFAR-100 data sets.

## F   POWER-LAW AND TRUNCATED POWER-LAW FIT

In this section, we show power law and truncated power law fitting results on all combinations of datasets and models. Fig. 22, Fig. 23, Fig. 24 show fitting results on CIFAR-10, Fig. 25, Fig. 26, Fig. 27 on CIFAR-100. As an example, we show the fitting results on the error profile of $\theta = 50\%$ in all figures in this section. In all combinations of datasets and models, while using more points gives higher accuracy and better prediction, both power-law and truncated power-law can get stable and precise prediction using a very limited number of small sample points.

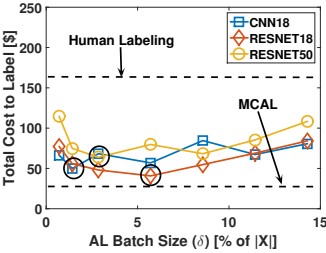
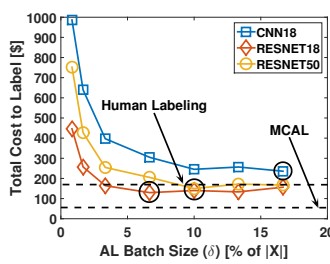
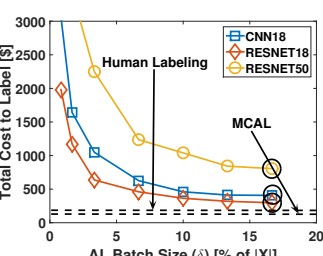

**FIGURE 16:** Performance of MCAL compared to active learning with different batch sizes $\delta$ on Fashion using Satyam labeling

**FIGURE 17:** Performance of MCAL compared to active learning with different batch sizes $\delta$ on CIFAR-10 using Satyam labeling

**FIGURE 18:** Performance of MCAL compared to active learning with different batch sizes $\delta$ on CIFAR-100 using Satyam labeling

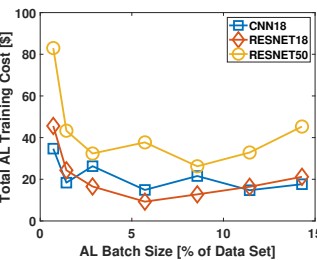
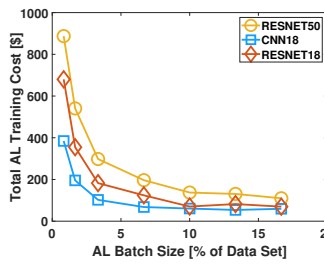
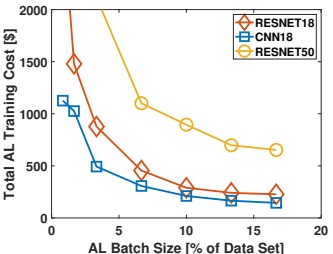

**FIGURE 19:** AL training cost (in \$) as a function of batch size ($\delta$) for Fashion Data Set using CNN18, RESNET18 and RESNET50.

**FIGURE 20:** AL training cost (in \$) as a function of batch size ($\delta$) for CIFAR-10 Data Set using CNN18, RESNET18 and RESNET50.

**FIGURE 21:** AL training cost (in \$) as a function of batch size ($\delta$) for CIFAR-100 Data Set using CNN18, RESNET18 and RESNET50.

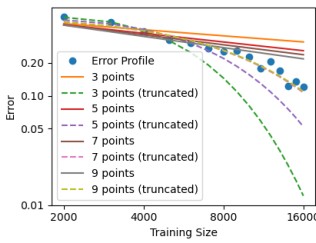
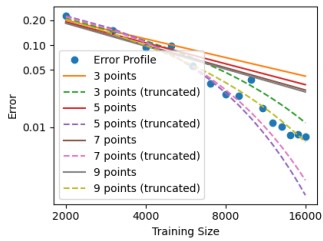
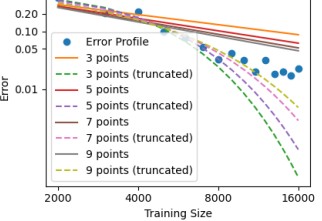

**FIGURE 22:** Power-law and Truncated Power-law fits on CIFAR-10 using CNN18

**FIGURE 23:** Power-law and Truncated Power-law fits on CIFAR-10 using RESNET18

**FIGURE 24:** Power-law and Truncated Power-law fits on CIFAR-10 using RESNET50

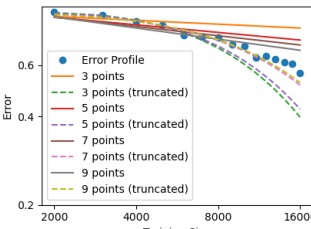
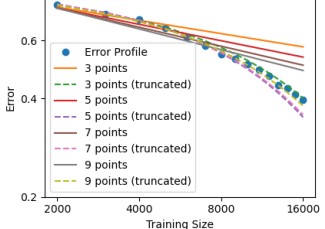
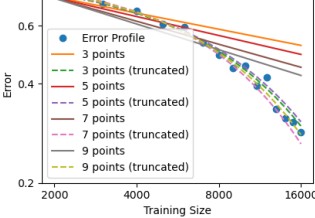

**FIGURE 25:** Power-law and Truncated Power-law fits on CIFAR-100 using CNN18

**FIGURE 26:** Power-law and Truncated Power-law fits on CIFAR-100 using RESNET18

**FIGURE 27:** Power-law and Truncated Power-law fits on CIFAR-100 using RESNET50

