# OpenReview forum: "MCAL: Minimum Cost Human-Machine Active Labeling"
_ICLR.cc/2023/Conference — ICLR 2023 poster_

### Official Review · Reviewer_hXWK · 2022-10-24

**Confidence:** 3
**Clarity, Quality, Novelty And Reproducibility:** As I mentioned above I didn't find th…
**Correctness:** 3
**Technical Novelty And Significance:** 2
**Empirical Novelty And Significance:** 2
**Recommendation:** 5

**Strength And Weaknesses:**

I generally have a negative impression about the paper. I can see that the approach is trying to
address a valid practical problem. The overall approach sounds very much like a common sense, i.e.
let's label some data points with already pre-trained model, and let human's label "interesting"
examples. The paper relies on standard metrics to assess what to give humans for labeling and how to
choose examples for automatic labels (i.e. examples where classifier is confident about the label).

The main technical contribution of the paper appears to be the use of the power law to predict how
many examples to label automatically (Sec. 3.1) and the overall system (Sec. 4).

After going through the paper several times, what I am left with is the following: The paper
presents an idea that sounds like a common sense, which is followed by hard to comprehend notation
and cost model (Sec. 2,3), followed by description of standard mechanisms of sample selection (Sec.
3.3), followed by hard to understand Sec. 4 that describes the algorithm.

The text does not communicate the ideas well unfortunately, and at least in my case it leaves the
reader unexcited about the approach. Perhaps using clear real-world examples would help here.

For example, I don't understand why power law in Sec. 4 even applies to the approach presented in
the paper. I can see how such power law works for increasingly larger training sets labeled by
humans. However, why does this law apply to automatically labeled examples, where adding larger
number of examples would result in adding more of the incorrectly labeled examples.

I addition I am not taking it for granted that "auto-labeling" generally works. Why is automatically
labeling examples where classifier is already confident going to help with classifying examples
where classifier is not confident? I understand that it might be the case, but I disagree that this
can be taken for granted.

Prior to reading this paper I believed that the main questions for the type of approaches proposed
in the paper is how to choose what to label manually, and how to choose what to label automatically.
If we knew how to do that well we would be in good shape to construct such systems. The paper does
not contribute to addressing these questions, and the paper didn't convince me that it makes other
valuable contributions.

I found some statements in the paper which I would like to have clarified:
1. "it chooses to label CIFAR-100 mostly using humans",
I am very curious why this is the case, and why similar outcome does not happen for ImageNet.

2. "MCAL’s goal is to select samples to train a classifier that can machine label the largest number
of samples possible."
That is certainly a good goal, but I do not understand how the approach achieves that. All I see in
the text is some application of the power law, whereas the problem above appears to be too complex to be
addressed by a power law.

3. "When MCAL uses margin or least confidence for L(.), the samples selected have high accuracy (close to 100%, Fig. 5)
. The samples selected by a core-set based algorithm such as k-centers is poorly correlated
with accuracy (Fig. 5) and margin (Fig. 6)."
I din't quite understand what is meant by these sentences. Why is accuracy relevant here if we are
selecting examples to be labeled and to be used for training of the next-iteration classifiers?
Also, if we select examples that have high confidence already why would such examples improve the
classifier?

Finally, I also doubt that the topic of the paper is within the scope of ICLR. For me ICLR stands
for interesting new representations (i.e. neural network models), or perhaps interesting techniques
for training neural networks. This paper sounds like a system paper that would perhaps be a good fit
for WACV (Winter Conference on Applications of Computer Vision) or some other more applied
conference (e.g. International Conference on Computer Vision Systems).

Positive:
The overall observation that in addition to human labeling cost there is also a computational cost is
valuable.


**Summary Of The Paper:**

The paper proposes a system for training of machine learning classification models. The main focus
of the paper is to reduce the overal cost of training. The total cost considered in the paper is a
combination of the cost due to human labeling and the cost due to computation (e.g. renting a GPU on
Amazon Cloud).

In order to reduce the training cost the paper proposes to rely on combination of active learning and
automatic labeling.

The training in the proposed MCAL approach proceeds iteratively. On each iteration the approach
selects which examples should be labeled by human annotators, and which examples are assigned labels
automatically using a classifier trained on previously labeled data (using automatic + human labels).

**Summary Of The Review:**

I refer to my text in the "Strength And Weaknesses" section. I would like to avoid repeating myself here.

---

> ### Author Response · Authors · 2022-11-18
> **Response to Reviewer 3**
>
> Thank you for your comments and feedback! We hope we can clarify the paper and improve its understanding in response to the comments. We will also revise the paper to add more intuitive explanations instead of math notations to hopefully improve the readability.
>
> - Algorithm clarification
>
> The usage of power law seems to be misunderstood. We use it only to estimate model accuracy for a fixed size of remaining data samples with increasing training sample size. It does not apply to errors with increasing machine-label sizes; MCAL never proposes to use power law on the latter. The key contribution is exactly what the reviewer expected “prior to reading the paper”: solving the problem of figuring out what to machine label and what to human-label and how much to allocate for each. It is very challenging to solve this problem without any information about the dataset. MCAL takes a “probing approach”: human label a small portion to train a model and estimate how it would perform at what training size. Figuring out the optimal tradeoff point in splitting the dataset for human and machine labeling is the key contribution of the paper. Also, machine labels are not guaranteed to be accurate, our evaluations using public dataset only validates its performance against human labels. But obtaining these data using a “less-performant model” before having these data to train a “perfect model” is rather crucial in operationalizing ML. It paves the way to train such a perfect model.
>
> - MCAL behavior on CIFAR100 vs ImageNet
>
> We caught this too! And it is a very interesting observation for us as well. The reason is likely that CIFAR100 has only 600 samples for each class whereas ImageNet has roughly around 800-1200 samples per class (CIFAR10, 6k samples per class). The headroom for machine labeling, after using some of the samples to learn, is bigger for CIFAR10 and even Imagenet compared to CIFAR100. In general, the observation is that it is very challenging for MCAL to use machine labeling to save cost if the samples are not enough for the model to achieve a decent error rate.
>
> - How does MCAL achieve machine labeling the most samples using the power law at a given error rate.
>
> We would love to clarify as this is the key contribution of the paper. The power law is only part of the algorithm, as it only helps estimating the model accuracy with respect to the active learning accumulated training size. The other important contribution is to apply this estimated accuracy to help the decision of choosing how much of the data should be machine labeled. MCAL estimates one such power law curve for each fixed sizes (theta) of remaining data samples from the dataset. Among the sizes, it filters for “valid” thetas, of which the model error rate must be below specified threshold (e.g. 0.05). For each filtered “valid” size theta, it also computes the corresponding human labeling cost (for 1-theta) and training cost using (1-theta) human labeled samples. Adding all the cost together, MCAL chooses the size theta that has the least total cost.
>
> - Comment on Fig 5 and Fig 6, and sample selection vs model performance
>
> There appears to be a misunderstanding which we would love to clarify. To begin with, the samples selected for training follow standard active learning mechanisms, i.e. if using the margin metric, samples with **lowest** margin (least confidence) are selected for human labeling and used in the next batch training. The sole purpose of Figure 5 is to show the correlation between sampling metrics and model accuracy, indicating that not all metrics are reflective of the model predictions. Margin and confidence can rank samples by accuracy, whereas entropy shows less correlation, and min-kcenter distance cannot rank model accuracy on samples at all. The sole purpose of Figure 6 is to show the correlation between different sampling metrics, or rather a zoom-in look at the distributions of all samples with respect to their metric values. Both Fig 5 and Fig 6 are meant to provide deeper insight on sampling metrics. They do not alter the way we do active learning.
>
> Thank you again for your valuable feedback! Hope our response clarifies most of the questions and helps communicate the key ideas of the paper. We are happy to respond to additional follow up questions.

---

### Official Review · Reviewer_9gqM · 2022-10-24

**Confidence:** 4
**Correctness:** 3
**Technical Novelty And Significance:** 3
**Empirical Novelty And Significance:** 3
**Recommendation:** 6

**Clarity, Quality, Novelty And Reproducibility:**

The proposed solution is simple but intuitive - including the model training objective under the AL task can be intuitively thought of being more "intelligent" about the selected samples while adhering to the realities of increasingly costly DL models. While the results are impressive, the clarity of presentation could be improved upon. This will also help with reproducing the results across domains.



**Details Of Ethics Concerns:**

While there are no direct ethical concerns, there may be unintended consequences in sample selection in proposed method - e.g does the algorithm provide equal exposure to labels for protected sub-groups for human labeling.



**Strength And Weaknesses:**

Some of the key strengths of the paper are as follows:

- The authors presented a general solution to a very important problem. With the impressive performance of AI/DL models for various tasks, the proposed method could be very important in operationalizing AI for real-world problems. Further, the MCAL Algorithm is not dependent on the underlying DL architecture / task and thus could be applied as a meta learner over multiple problems
- The authors have presented evaluations across multiple datasets/publicly available benchmarks to support the claim about the effectiveness of the solution
- Further, the detailed analysis about the reduction in labeling cost from multiple view points provides better insights for the method to be operationalized

The paper could improve upon the following aspects

- The presentation of the paper can be improved upon. The two components of MCAL algorithm is hard to follow. Symbols are sometime introduced before they are properly defined (e.g. delta in section 3.1, 3rd line from end of the section; first introduced in 3.2) and intuitions behind certain selections not presented well
- While the authors provide interesting insights about the drivers of the reduced cost, the discussion is lacking depth when considering the problem in a holistic manner. For example, in addition to labeling cost, other metrics of interest can include turnover time which can be affected as part of this multi-loop process
- In a similar manner, more details/analysis on which samples are selected during the AL phase would be useful. The authors presented a comparative analysis of the sample selection candidates - however, the intuition and effect of the selected samples may need further analysis. This could also be useful for downstream tasks in identifying "good enough" solutions under budget constraints

**Summary Of The Paper:**

The authors presents a novel active learning setup using a "hybrid human-machine labeling" framework to reduce the cost of labeling/annotating large datasets. They compared their approach on a number of public datasets and claims to provide a 6x overall reduction in cost to entire human labeling of the dataset.

**Summary Of The Review:**

Overall, the paper discusses an intuitive and simple solution to gather ground truths for large datasets and reduce the overall labeling costs for the process. While there are avenues for improving the paper, the results are impressive and general enough to have a significant impact.

Apart from the previously highlighted strengths and weakness, the authors may also want to address the following:

- Section 3.1, the intuition for the upper-truncated power-law could be motivated better. For example, a plot showing the effects of various values of the parameters from equation (3) could be useful to understand the desired shaping of the rewards
- MCAL Algorithm -provide better details about the fitting procedure. Discuss about the cost for the optimization function
- Provide intuitions behind the sample selection - e.g. how does the ranking of samples selected change with reduced labeling cost etc. Investigate possible issues with fairness / group fairness in selection of samples
- In recent years, a number of papers have identified labeling/data issues in the original benchmarks. The authors may want to compare if their proposed solution is able to circumvent those issues

---

> ### Author Response · Authors · 2022-11-18
> **Response to Reviewer 2**
>
> Thank you for your valuable insight! We appreciate it. Indeed data is increasingly becoming the pain point of operationalizing ML and scaling MLOps to various applications.
>
> - Presentation
>
> Thank you for pointing out this issue. We have carefully gone through the notations again to make sure symbols are introduced before usage. And we have improved the description of the core two modules, adding more explanation before symbol usage.
>
> - Metrics other than labeling cost including turn-around time
>
> While the prohibitive labeling cost is the major pain point in scaling MLOps, indeed we agree that time to turn-around labeled samples and other metrics could be important factors depending on the application. In fact, in our experience crowd-sourcing the labeling of a large dataset, the turn-around time is highly variable and sometimes much longer than training and machine labeling time. We should be able to accommodate turn-around time into the optimization, but for this to work well, would need robust estimates of the likely turn-around for a given number of samples. We can add a brief discussion of this if space permits.
>
> - More details on sample selection
>
> This is a very nice idea. We will add a few examples of samples selected in the appendix.
>
> - Effect of power-law parameters in equation 3
>
> Figure 3 shows power-law fitting curves with various parameters estimated from different numbers of points. We have also included, in the appendix, Figure 22-27 showing different parameter estimations on different model and dataset pairs.
>
> - Power law fitting details
>
> We have used curve_fit from the scipy package to fit the power-law curves. We will release the estimation and iteration algorithms source code.
>
> - Intuition behind sample selection, impact on cost
>
> This is a very interesting topic. Sampling metrics play a significant role in MCAL. Figure 6 alludes to this a little bit. Different metrics rank samples differently, and therefore impact the training performance and then effect on the total cost. In our experiments, we see margin perform really well, confident and entropy relatively well, while min k-center distance almost brings no cost reduction.
>
> - Labeling issues with public benchmarks
>
> We have observed similar issues by resending public data out to labeling services. It is possible, as mentioned in response to reviewer Dg4t, to extend MCAL to leverage the sampling metrics in the framework, to mark irregular/wrong human labels, or out-of-distribution samples, in an automated fashion. For example, an incorrectly labeled sample may have very different sampling metric values compared to correctly labeled majority samplers. This is a very interesting direction to explore next.
>
> Thank you again for your valuable feedback!

---

### Official Review · Reviewer_Dg4t · 2022-10-26

**Confidence:** 3
**Correctness:** 3
**Technical Novelty And Significance:** 2
**Empirical Novelty And Significance:** 3
**Recommendation:** 8

**Clarity, Quality, Novelty And Reproducibility:**

The paper presents very interesting approach but there is no comparison with other existing cost sensitive active labeling methods (expect with active learning method).  A comparison to other weak labeling methods with active learning would have made a good comparative study in this work.


**Strength And Weaknesses:**

Strength:
- The paper presents approach that allows control over accuracy and cost with the cost minimization objective to launch data labeling task for developing a high accuracy model training.
- The approach scales well with complexity of the data and also suggests (with a ‘exploration tax’) if it’s feasible to automatically label the data or not (given the size and complexity)
- The approach can use any active learning metric for selecting samples for human-labeling. Also they found empirically that some popular metrics (like coreset) don’t work as well as others with this objective.
- The approach is extensible to allow multiple model architectures

Weakness:
- It is not clear if the approach would work with a dataset containing imbalanced distributions and/or open classes to decide which ones to human-level vs machine-label. Also not clear if the power-law distribution would follow in the situation.
- Fig. 4 needs more explanation - the role of batch size for model errors is low which is unintuitive.
- The human-labeling is assumed to be oracle in this work - how realistic is this assumption - if the model can accommodate imperfect oracles in human labels, the method would become more realistic.
- Fig. 6 needs more explanation
- It is not clear how does ‘fast search’ work in the method.


**Summary Of The Paper:**

The paper presents an ‘active labeling’ (inspired from active learning) based approach to automatically label parts of a dataset given some error bounds by jointly optimizing for human labeling and machine labeling costs (eg. training costs) together. It formulates the problem as a minimum-cost labeling problem in an optimization framework that minimizes total cost by jointly selecting which samples to human label and which to machine label. A major assumption of this work is that it considers the truncated power-law distribution to mimic how model error behaves with training set size.

**Summary Of The Review:**

This paper presents some practical ideas for cost reduction for data labeling which is growing need today. The work presents a solid extensible framework where one can set various thresholds for performing the labeling tasks. The methods have been testing on multiple datasets and understanding the role of various parameters have been explored and discussed well in the paper.

---

> ### Author Response · Authors · 2022-11-18
> **Response to Reviewer 1**
>
> Thank you for your valuable feedback and comments! We really appreciate it.
>
> - Dataset containing imbalanced distribution and/or open classes.
>
> While MCAL works best for reasonably balanced datasets, we can extend it to support dataset imbalance while keeping overall cost low. We can exploit active learning metrics to craft out-of-distribution detectors, and “push” the imbalanced samples more towards human labeling, thus keeping the majority of balanced data for machine labeling. For open classes, if it’s a minority of the dataset, it could also benefit from such out-of-distribution detectors. If it’s a major class, it would make sense for the labeling model to include a “default/unknown” class output, and the MCAL framework could treat them as a regular class just with “unknown” labels. There may be other approaches, and we will explore them in the future.
>
> - Explanation on of Figure 4
>
> As the reviewer correctly points out, model accuracy depends on batch sizes during active learning. With smaller batch sizes, model accuracy grows faster as total training size |B| grows. Fig 4 shows the accuracy difference when the total training samples is |B| = 16000, *towards the end of the accuracy vs training size curve*. At this point, accuracy converges to the maximum model accuracy, which is limited by model capacity, not batch size. Oftentimes, this is where the model can be faithfully applied to machine-label most of the rest of the data to achieve low total cost.
>
> - Human-labeling-as-oracle assumption.
>
> Indeed, it would be nice to accommodate human labeling errors in MCAL. In theory, if we can characterize the error rate of human labeling, we can account for human errors in making the cost decisions (in much the same way as we take the error of machine labeling into account). The real problem is that we lack even an empirical characterization of this error rate.
>
>  - Explanation of Figure 6
>
> Figure 6 compares different sampling metrics for each data sample. The X-axis shows the margin metric, each dot in the plot shows a data sample evaluated by one metric (confidence as green, entropy as orange, and min k-center distance as blue). It shows that confidence and margin are positively correlated, entropy and margin are negatively correlated, but min-kcenter distance is almost not correlated (evenly distributed) with margin at all. This figure shows how each metric would sample the data in the active learning process. It also gives some insight as to why each metric performs differently end-to-end.
>
> - Comment about “fast search”
>
> For lack of a better word, we use fast search to describe the process of exploring the next data point on the power-law estimation curve (as shown in figure 3), and the corresponding number of samples to be labeled by humans. Instead of using fixed batch sizes, “fast search” efficiently and economically accumulates data points on the curve to refine the power-law estimation parameter and reach the final lowest cost point faster.
>
> Thank you again for the feedback. We have revised the paper to address the above comments.

---

### Decision · Program_Chairs · 2023-01-20

**Decision:**

Accept: poster

**Justification For Why Not Higher Score:**

Presentation lacks clarity and the experimental settings are not realistic to judge the immediate impact of this work.

**Justification For Why Not Lower Score:**

The problem setting studied in this paper is important, and the solution is technically correct and validated by experiments.

**Metareview: Summary, Strengths And Weaknesses:**

*Summary*: The paper studies the problem of active labeling of image datasets. The authors propose a minimum cost optimization framework that jointly models the cost of human labeling and machine labeling. The framework is evaluated on a number of image classification datasets.


*Strengths*: (1) Active labeling is an important topic for current ML systems, given that they are trained on massive amounts of supervised and self-supervised data. (2) The framework seems agnostic to the underlying model and the labeling task. This makes it more broadly useful beyond the settings used in the paper. (3) The paper has good empirical evaluations, and includes findings about how current metrics may not work well for active labeling.

*Weaknesses*: (1) The presentation in this work is quite unclear. This point was raised by all the reviewers as well. (2) The empirical evaluations in this paper are limited: the unlabeled data is "close" world, and not heavy tailed. This makes the overall applicability of the paper weaker.

**Note From Pc:**

if the above contains the word "oral" or "spotlight" please see: "oral" presentation means -> notable-top-5% and "spotlight" means -> notable-top-25%. As stated in our emails, we are disassociating presentation type from AC recommendations